# Interactive Effects of Intraspecific Competition and Drought on Stomatal Conductance and Hormone Concentrations in Different Tomato Genotypes

Yang Gao [1,*](id), Yueping Liang [1], Yuanyuan Fu [2], Zhuanyun Si [1] and Abdoul Kader Mounkaila Hamani [1]

[1] Key Laboratory of Crop Water Use and Regulation of Ministry of Agriculture and Rural Affairs, Farmland Irrigation Research Institute, Chinese Academy of Agricultural Sciences, Xinxiang 453002, China; liangyueping@caas.cn (Y.L.); sizhuanyun@caas.cn (Z.S.); 2020y90100004@caas.cn (A.K.M.H.)

[2] College of Plant Science, Tarim University, Alar 843300, China; fyy2016060105@163.com

[*] Correspondence: gaoyang@caas.cn

**Abstract:** Plant physiological responses to various stresses are characterized by interaction and coupling, while the intrinsic mechanism remains unclear. The effects of intraspecific competition on plant growth, stomatal opening, and hormone concentrations were investigated with three tomato genotypes (WT-wild type, Ailsa Craig; FL-a abscisic acid (ABA) deficient mutant, *flacca*; NR-a partially ethylene-insensitive genotype) under two water regimes (full irrigation, irrigation amount = daily transpiration; deficit irrigation, 60% of irrigation amount in full irrigation) in this study. Three kinds of competitions were designed, i.e., root and canopy competition, non-root competition, and non-canopy competition, respectively. Intraspecific competition reduced plant leaf area and stomatal conductance ($g_s$) of wild-type tomato, accompanied by ABA accumulation and ethylene evolution. Intraspecific competition-induced decrease in $g_s$ was absent in FL and NR, indicating ABA and ethylene involved in plant response to intraspecific competition. As soil water becomes dry, the competition decreased $g_s$ by elevating ABA and ethylene accumulations. Under severe drought, the competition-induced decline in $g_s$ was covered by the severe drought-induced decrease in $g_s$, as hydraulic signals most probably dominate. The absence of canopy competition insignificantly influenced plant stomatal opening of well-watered tomato, as canopy separation minimized the plant neighbor sensing by ethylene and other signals. Whereas under water deficit condition, the absence of canopy competition significantly reduced ABA accumulation in roots and then stomatal conductance, indicating the belowground neighbor detection signals maybe enhanced by soil drought. The absence of root competition increased ethylene evolution, confirming the importance of ethylene in neighbor detection and plant response to environmental stress.

**Keywords:** intraspecific competition; tomato; plant hormones; above- and belowground competition

## 1. Introduction

Both biotic and abiotic stresses affect plants normal growth and development, and significantly decrease their productivity [1]. Abiotic stress includes drought, salinity, flood, etc., and biotic stress includes attack by various pathogens, and competition from neighbors. As plants responses to these stresses are complex, more attention is focused on plants responses to particular abiotic or biotic stress [2]. Phytohormones play central roles in sensing biotic and abiotic stresses [3–8]. Moreover, physiological effects of signal response to various stresses are characterized by interaction and coupling, while the intrinsic mechanism is still unclear.

Plant senses the water deficiency signal and initiate physiology response to the drought signal [9]. Abscisic acid (ABA) plays an important role in plant responses to drought and optimizes water use. The ABA pathway has been used to improve water use efficiency and drought resistance [10]. ABA can induce the closure of stomata together with several

environmental factors. Pantin et al. [11] proposed the dual effect of ABA on stomatal closure, i.e., the biochemical effect on guard cells and the indirect hydraulic effect on water permeability within leaf vascular tissues. Ethylene, a gaseous phytohormone, regulates both growth and senescence. There were contradictory claims on the effects of ethylene on stomatal opening [12]. Pierik et al. [13] suggested that the effects of ethylene on stomatal conductance depended on ethylene production and sensitivity. Moreover, the interactive role of ethylene and ABA as well as other hormones has been proved in activating the phytohormones regulation of several processes [14–16].

Inter-/intra-specific competition for growing space and limited resources and drought are the important biotic and abiotic factors inhibiting plant growth [7,8]. Plants can detect neighbors by multiple hormones and respond to them in various ways [3,17–19]. Over-representation of phytohormone-responsive genes was observed in competing Arabidopsis plants, confirming the competing-induced involvement of plant hormones [20]. Ethylene is an important hormone in determining plant responses to neighbors, such as shoot elongation and leaf and stomatal movements [7,21,22]. Pierik et al. [17] found that the ethylene-insensitive transgenic tobacco could reduce shade avoidance responses to the neighbors. Ethylene generally maintains stomata at sub-maximum apertures despite the relatively non-stressful conditions [7]. Vysotskaya et al. [7] explained that the decline in stomatal conductance induced by the neighborhood was due to the increased ethylene production in competing plants. ABA, an essential hormone adjusting stomatal opening, is also involved in plant's response to neighborhood [7].

Apart from the effect of ABA on stomatal closure, little attention was paid to the interaction between hormones [23,24]. As ethylene has an opposing effect on the stomata by inhibiting ABA-induced stomatal closure [25], more experiments are necessary to investigate the co-regulation of ABA and ethylene in stomatal responses to the presence of neighbors. Moreover, recent researches have indicated that cytokinin and auxin participate in plant adaptation to competition [3,26]. As response of hormones to stressful environments is a complex signaling network, the mechanism of phytohormone's regulation in plant adaptations to stresses has not been well elucidated. Under the condition of low population density, the inter-/intra- specific competition was mainly observed belowground [27]. However, limitation in aboveground space or belowground resources in high population environments may favor different suites of plant traits [26]. Despite the interdependence of above- and belowground competition [28], no studies have addressed the effects of below- and aboveground interaction between different species on plant's response to environmental stresses, when grown in a mixture under both abiotic and biotic stresses conditions. Stomatal opening is a fundamental response of plants to the environment, regulating carbon gain and water loss [7,29,30]. A signaling network of hormones controlling stomatal movement has been well established [31–34], while little information is available on the stomatal response to the presence of neighbors, more specifically, the above- and belowground completion. Therefore, the objectives of this study were to investigate the interactive effects of intraspecific competition and drought on plant growth and stomatal response in different tomato genotypes, and to analyze the influence of aboveground and belowground competition on plant hormones accumulation (ABA and ethylene) and stomatal opening.

## 2. Materials and Methods

### 2.1. Plant Materials

Tomato (*Solanum lycopersicum* L.) was used as a model species. The wild-type (WT) of tomato was Ailsa Craig. The Never-ripe mutant (NR) was the partially ethylene-insensitive genotype, and the *flacca* mutant (FL) was the abscisic acid (ABA)-deficient tomato mutant. Seeds from the three genotypes obtained from the Tomato Genetics Resource Center (University of California, Davis, USA), were germinated in compost (John Innes No. 2) and covered with black plastic. After 6–7 days, the plastic was removed to prevent etiolation of the seedlings. After a further 8 days, seedlings were transferred to pots filled with the same

substrate, and grown in a walk-in controlled environment room with a day/night temperature of 32/16 °C and a 12 h photo-period (06:00–18:00). Light intensity at plant height was between 400 and 600 µ mol m$^{-2}$ s$^{-1}$ PPFD (Photosynthetic Photon Flux Density).

## 2.2. Experimental Design

Two irrigation levels were designed as full irrigation (irrigation amount = daily transpiration) and deficit irrigation (60% of irrigation amount in full irrigation). Three kinds of competition were designed, i.e., root and canopy competition, non-root competition, and non-canopy competition, respectively. There were three sub-treatments (two plants in one pot) in each competition, i.e., WT/WT, WT/NR, and WT/FL, respectively. CK was a single plant of the three species in one pot. Two different size pots were used to treat a single plant, i.e., 1.86 L and 0.94 L, respectively. The pot size for the competing plants was 1.86 L. Therefore, 30 treatments with 10 replications were carried out in this experiment (Table 1). Each treatment was replicated twice. The pot in the treatment without root competition was completely separated into two equal parts using an acrylic divider, glued to the inner wall and bottom of pot. For the treatment without canopy competition, a transparent glass barrier was placed between the aerial portions of two plants to totally separate the shoot components.

**Table 1.** Experimental design in this study. WT-wild type, NR-never ripe mutant, FL-*flacca* mutant.

| Irrigation Factor | Competition Factor | | |
|---|---|---|---|
| Full Irrigation | Single Plant | $^a$ 1.86 L | WT, NR, FL |
| | | $^a$ 0.94 L | WT, NR, FL |
| | Competing Plants | With root and canopy competition | WT/WT, WT/NR, WT/FL |
| | | Without root competition | WT/WT, WT/NR, WT/FL |
| | | Without canopy competition | WT/WT, WT/NR, WT/FL |
| Deficit Irrigation | Single Plant | $^a$ 1.86 L | WT, NR, FL |
| | | $^a$ 0.94 L | WT, NR, FL |
| | Competing Plants | With root and canopy competition | WT/WT, WT/NR, WT/FL |
| | | Without root competition | WT/WT, WT/NR, WT/FL |
| | | Without canopy competition | WT/WT, WT/NR, WT/FL |

$^a$ The volume of pots.

## 2.3. Plant Measurements

At 10 days after transferring tomato plants into pots, plants were harvested to measure leaf area using a leaf area meter (Licor Model 3100 Area Meter, Cambridge, UK). The dry weight of the leaf, stem, and root was also measured. Leaf water potential (LWP) of tissue discs from mature leaves of tomato plants was measured with Wescor 5100 thermocouple psychrometers (Logan, UT, USA). The stomatal conductance ($g_s$) was measured between 10:00 and 10:30 using a porometer (AP4, Delta-T Devices Ltd., Cambridge, UK). In some experiments, the soil surface was covered with aluminum foil to prevent water evaporation, and then the plant transpiration was determined.

## 2.4. Plant Hormone Analysis

Bulk leaf ABA concentration and root ABA concentration were measured with a radioimmunoassay (RIA) using the monoclonal antibody AFRC MAC 252 [35]. The youngest and fully expanded leaflet was harvested for ABA measurement. Plants were sampled simultaneously (10:00–10:30) on each harvesting day to avoid diurnal effects on foliar ABA concentration. Leaflets and roots (on the same plant) were sampled, snap frozen in liquid nitrogen, freeze-dried for 48 h, and then finely ground. A small number of samples (10–15 mg dry weight for leaflets, and 30–40 mg dry weight for roots) was needed for ABA analysis. Then, the samples were diluted with deionized, distilled water (1:70 for leaflets, and 1:25 for roots). Samples were then placed on a shaker in a cold room (4 °C) overnight to extract

ABA. A standard curve was determined with standards in a serial dilution of synthetic unlabeled (±)-cis, trans-ABA (Sigma Let., Dorset, UK). ABA concentration was calculated using the 'logit' transformation by referring to the standard curve after linearization.

To determine ethylene evolution rates, leaflets and roots on the same plant sampled for ABA determination were weighed and placed in 28 mL glass vials containing saturated filter paper, which was then sealed with a rubber puncture cap. The leaflets and roots samples were incubated for 60 min under a lamp (200 μ mol m$^{-2}$ s$^{-1}$) and in a dark chamber, respectively. A 1 mL headspace sample was withdrawn using a gas-tight syringe, then manually injected into a gas chromatograph (6890N, Agilent Technologies UK Ltd., Wokingham, UK; Networked GC system, method: Ethylene split. M, software: Enhanced Chemstation Online GC) equipped with a J&W HP-AL/S (50 m × 0.537 mm × 15.0 mm) column (HiChrom Ltd., Reading, UK). This was maintained for the first 5 min at 100 °C to resolve ethylene, and then ramped at 15 °C to 150 °C and held for 1.5 min to drive off any water vapor introduced onto the column by sample injection. The carrier gas was helium at a flow rate of 5.7 mL min$^{-1}$, and detection was by flame ionization. The rate of ethylene evolution was determined referring to the peak areas of the known ethylene standards (99.995% minimum purity, BOC Special Gases, Manchester, UK), and corrected for tissue FW and time in incubation.

### 2.5. Statistical Analysis

All data were analyzed by one-way analysis of variance (ANOVA). A less significant difference (LSD) and Student's *t*-test were carried out with SPSS21.

## 3. Results

### 3.1. Effects of Intraspecific Competition on Plant Growth and Stomatal Opening of WT Tomato

At 4 days after the competition, there was no significant difference in leaf area between the single and competing plants of WT tomato (Figure 1a,b). Subsequently, compared with the single plant, the competing averagely decreased leaf area by 21% under full irrigation and 26% under deficit irrigation. Transpiration was calculated per-unit leaf area basis [5]. The intraspecific competition significantly decreased transpiration during the sampling period, on average 33% and 28% lower than the single plant under full irrigation and deficit irrigation, respectively (Figure 1c,d).

Under full irrigation, the leaf water potential (LWP) of single WT plant (−0.58 ± 0.03 MPa) was similar to that of the competing plants (−0.60 ± 0.03 MPa). Furthermore, there was no significant difference in LWP between the single plant and competing plants under deficit irrigation (Figure 2a). The abaisial stomatal conductance (g$_s$) of fully expanding leaves are shown in Figure 2b,c. Under the well-watered condition, the intraspecific competition significantly reduced the g$_s$ of WT tomato. As soil moisture depleted gradually, the difference in g$_s$ between the single tomato and the competing plants was markedly decreased from 26% at 4 days after competition to 16% at 10 days after competition. Although the LWP was similar between the single plant and competing plants, the difference in g$_s$ between the two treatments was measured in our experiment. The result indicated that the non-hydraulic signals contributed for regulating the stomatal opening of competing plants. However, under severe drought (at 12 days after competition), the g$_s$ of single tomato was not significantly higher than that of competing plants, as the LWP in competing plants was lower than that in the single plant.

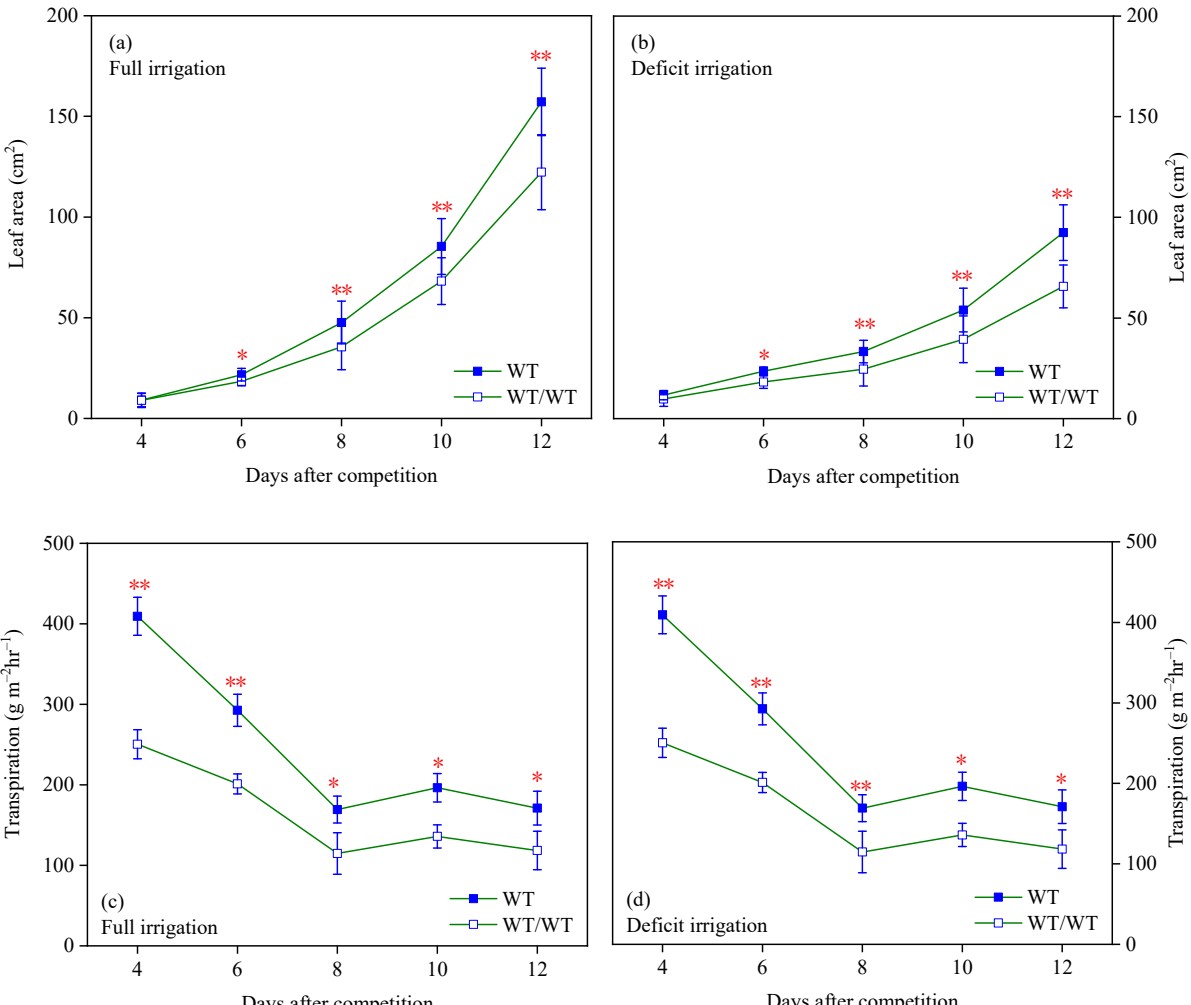

**Figure 1.** Variations of leaf area and transpiration in CK and the treatment of competition under full irrigation (**a**,**c**) and deficit irrigation (**b**,**d**). * Indicates significant difference between treatments (* $p < 0.05$, ** $p < 0.01$).

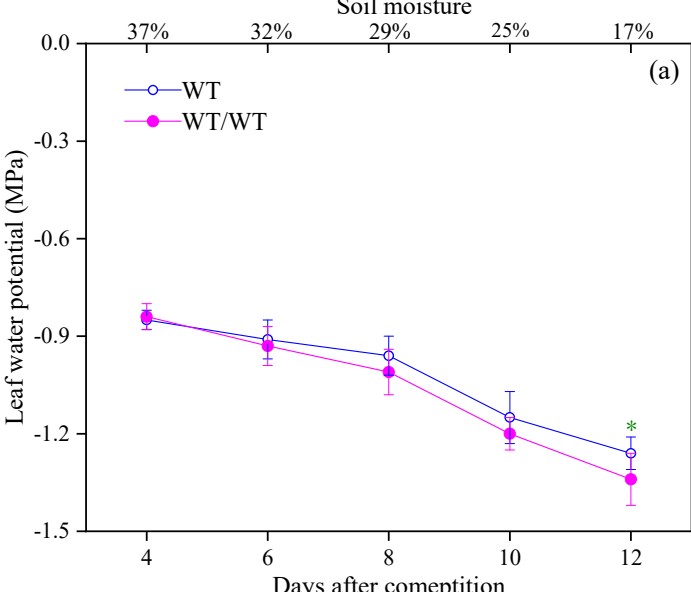

**Figure 2.** *Cont.*

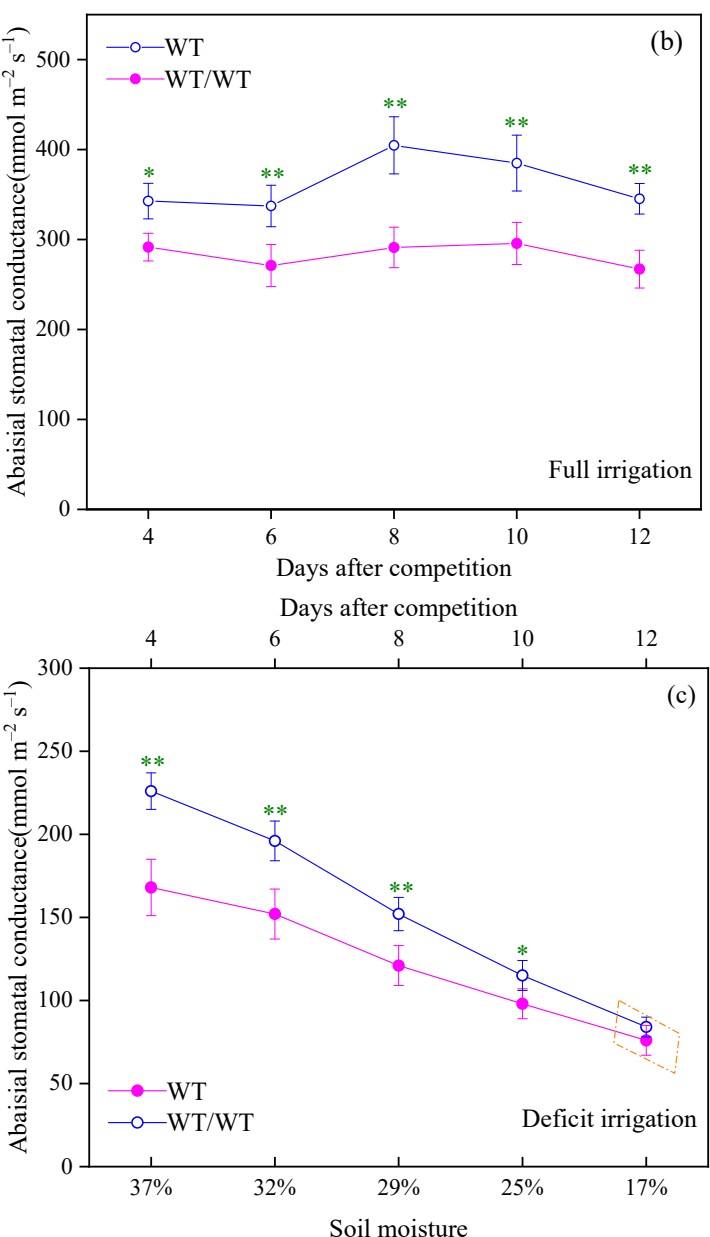

**Figure 2.** Variation of leaf water potential ((**a**) deficit irrigation), abaxial stomatal conductance ((**b**) full irrigation, (**c**) deficit irrigation) in the single plant and competing plants of wild type tomato (* $p < 0.05$, ** $p < 0.01$).

### 3.2. Involvement of ABA and Ethylene in Plant Response to Intraspecific Competition

As a mutant of ABA-deficient, in the WT/FL competing pot, the $g_s$ of well-watered FL plant ($686 \pm 24$ mmol m$^{-2}$ s$^{-1}$) was similar to the value ($681 \pm 18$ mmol m$^{-2}$ s$^{-1}$) of water-stressed FL tomato (Figure 3), while the $g_s$ of water-stressed WT tomato was markedly lower than the value of well-watered WT plant. Like the WT/WT competing pot, the intraspecific competition from the WT/FL competing pot significantly reduced the $g_s$ of well-watered WT. When soil moisture reduced from 36% to 26%, the difference in $g_s$ between the single WT tomato and the competing WT in the WT/FL competing pot was on average 16% ($p < 0.01$); this difference was not significant as soil moisture depleted below 20%. The difference response of stomatal opening to intraspecific competing between WT and FL indicated that the ABA involved in tomato responds to competition. ABA concentration in the single and competing WT tomatoes under the full and deficit irrigation is shown in Table 2. Whether under the well-watered or water deficit condition, the

intraspecific competition markedly increased the foliar ABA accumulation in WT tomato (Table 2). Under the full irrigation, the intraspecific competition significantly decreased the root ABA concentration, whereas increased ABA accumulation in roots under soil drought.

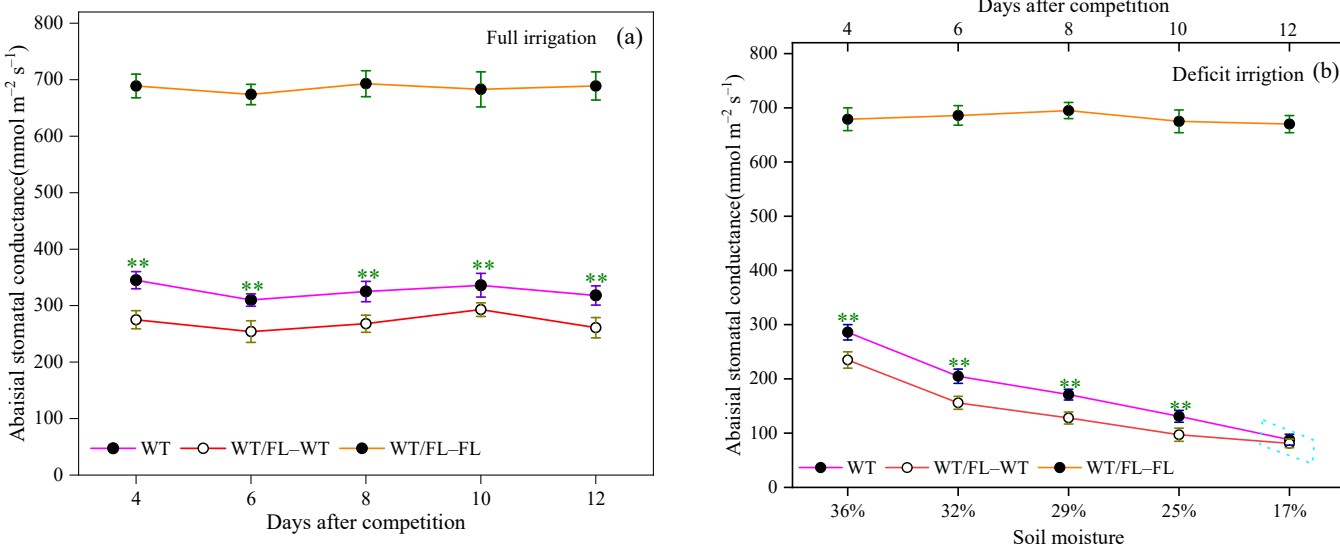

**Figure 3.** Variation of abaxial stomatal conductance ((**a**) full irrigation, (**b**) deficit irrigation) of FL and WT in competing pot. The wild-type tomato and *flacca* mutant was planted in one pot (WT/FL) (** $p < 0.01$).

**Table 2.** ABA concentration (ng g$^{-1}$ DW) in leaf and root of the single and competing WT tomato under full and deficit irrigation. Data are means ± standard error (SE). Different letters denote statistically significant differences between treatments ($p < 0.05$).

| Treatment | Deficit Irrigation | | Full Irrigation | |
|---|---|---|---|---|
| | Leaf | Root | Leaf | Root |
| WT | 961.5 ± 52 b | 115.4 ± 4.5 c | 332.6 ± 23 c | 65.4 ± 3.4 b |
| WT/WT-WT | 1080.4 ± 75 a | 155.3 ± 3.2 b | 505.2 ± 17 a | 43.5 ± 2.1 a |
| WT/FL-WT | 1136.6 ± 61 a | 174.7 ± 4.1 a | 458.6 ± 21 ab | 47.7 ± 3.1 a |

When growing WT and NR tomato in a pot, whether under full or deficit irrigation, the intraspecific competition did not reduce the NR plant's leaf area, transpiration, and abaxial g$_s$ (Figure 4). In the WT/NR competing pot, the intraspecific competition had more significant influence on the g$_s$ of WT tomato (Figure 5a). Under the full irrigation, the g$_s$ of WT plant was averaging lower than that of NR plant by 22%. The difference in g$_s$ between the WT and NR tomato gradually decreased from 25% to 7% with a decrease in soil moisture. Moreover, this difference was not significant as soil moisture was lower than 28%. Overall, the distinct response to intraspecific competition between WT and NR tomato indicated that ethylene contributed to tomato response to intraspecific competition.

In the WT/NR competing pot under full irrigation, the foliar accumulation of ABA in competing WT plants (WT-C) was significantly greater than that in the single WT plant by 39% and 44%, respectively. In contrast, the single WT plant had a higher ABA concentration in roots, 35% and 23% greater than that in the competing plants (WT-C), respectively (Figure 5b). Under water deficit condition, the foliar accumulation of ABA in the competing plants was significantly greater than that in the single WT plant, and the value in the competing NR tomato was significantly higher (35%) than that in the competing WT plant. However, there was no significant difference in root ABA accumulation between the competing plants and the single plant under water deficit condition. The effects of intraspecific competition on foliar ethylene evolution between the single WT and competing

WT and NR tomato are shown in Figure 5c. Under the well-watered condition, the foliar ethylene evolution in the competing NR and WT tomatoes was 73% and 28% higher than that in the single WT plant. The value in the competing WT tomato was significantly greater than the competing NR plant by 26%. There was no significant difference in foliar ethylene between the competing plants and the single WT plant under deficit irrigation.

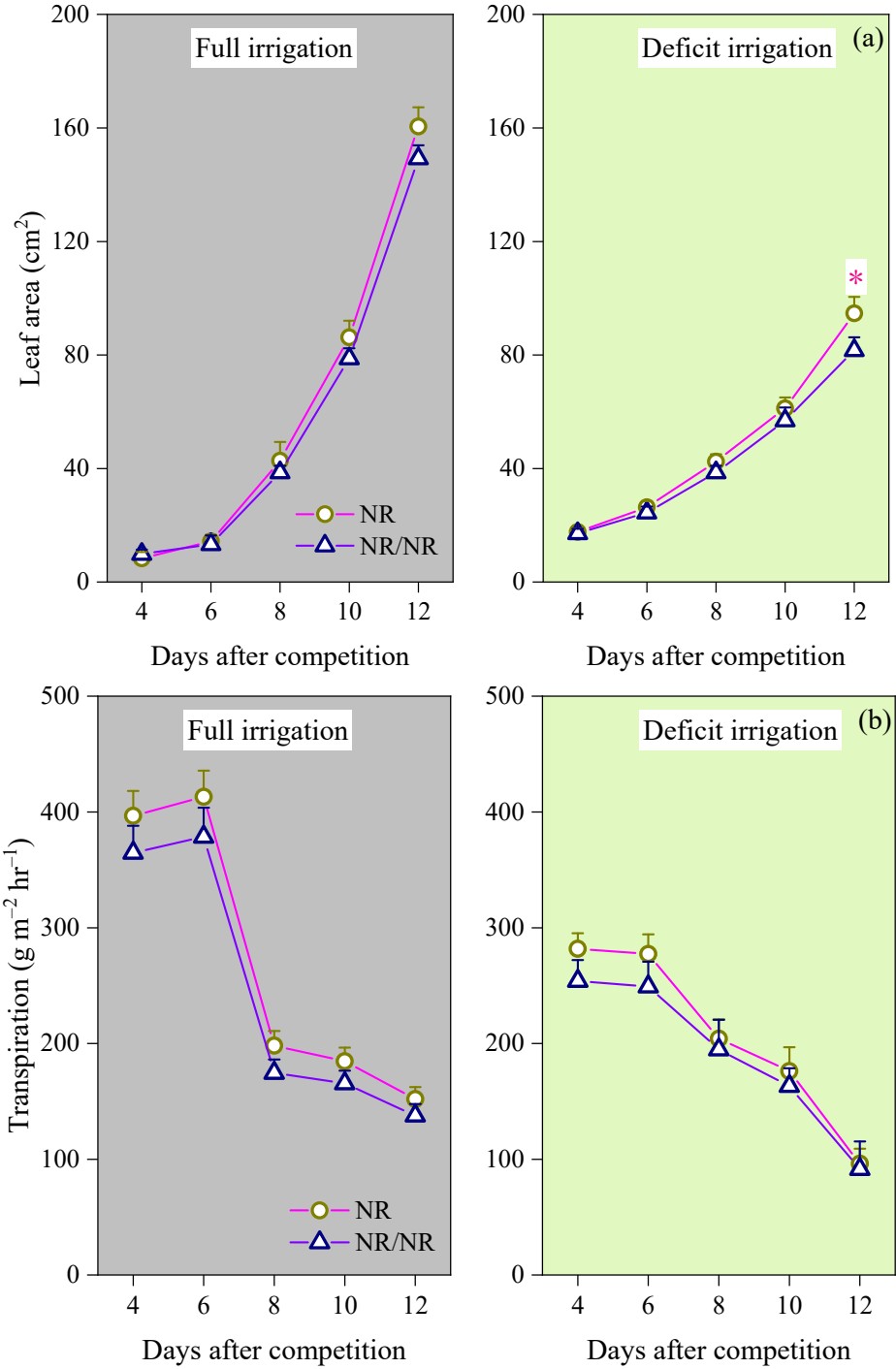

**Figure 4.** *Cont.*

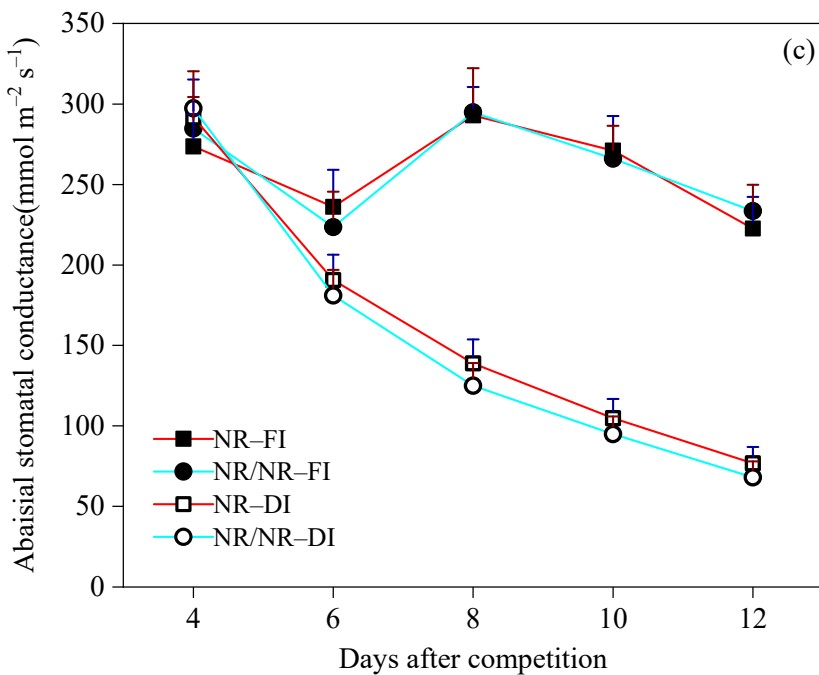

**Figure 4.** Variation of leaf area (**a**), transpiration (**b**), and abaxial stomatal conductance (**c**) of the single and competing plant of NR tomato under full irrigation and deficit irrigation (* $p < 0.05$).

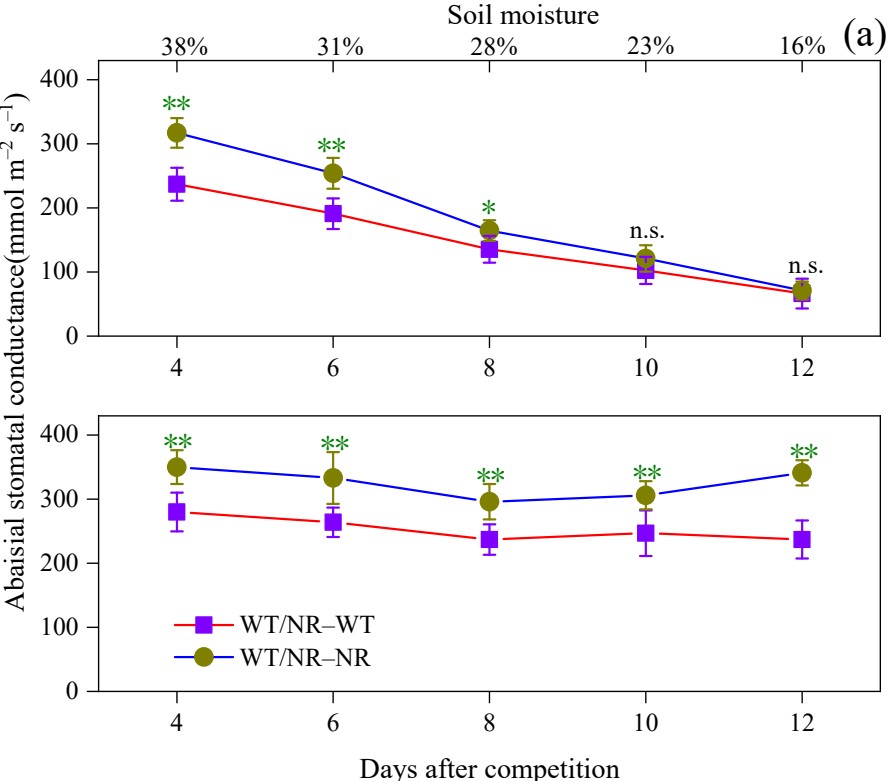

**Figure 5.** *Cont.*

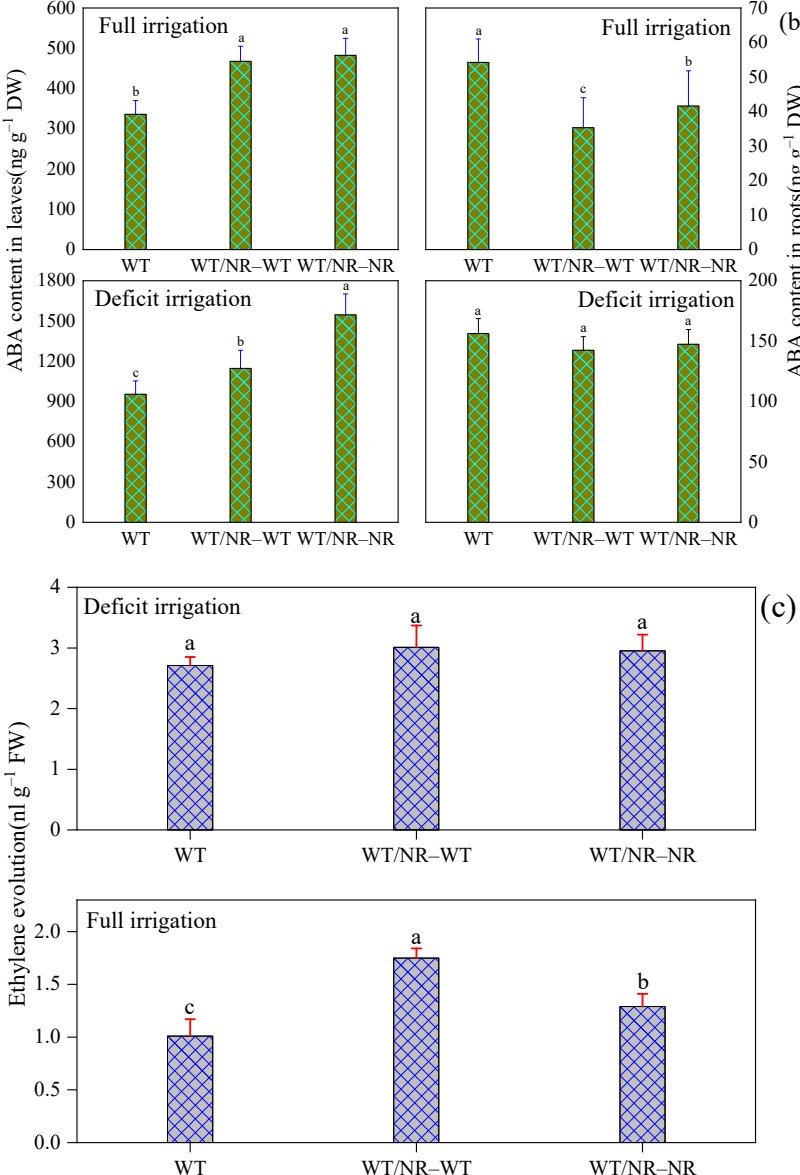

**Figure 5.** Effects of intraspecific competition on stomatal conductance (**a**), ABA content in leaves and roots (**b**), and ethylene evolution (**c**) of the single WT tomato and the competing plants from pot with WT and NR tomato under full irrigation and deficit irrigation (* $p < 0.05$, ** $p < 0.01$). Different letters denote statistically significant differences between treatments ($p < 0.05$). n.s. means statistically insignificant difference between treatments ($p > 0.05$).

*3.3. Effects of Above- and BelowGround Competition on Plant Response to Competition*

With sufficient water supply, the pot size had insignificant influences on plant growth, stomatal opening, and hormone synthesis of the competing tomato (data not shown). From the first day to 10 days after deficit irrigation, soil moisture in the small pot with a volume of 0.94 L was slightly lower than that in the big pot with a volume of 1.86 L, and the difference was not significant. While at 12 days after deficit irrigation, the value in the big pot was 17.3%, significantly greater than 13.0% in the small pot. At 10 days after deficit irrigation, there were insignificant differences in leaf area, transpiration, and foliar ABA concentration of tomato between the small pot and big pot. In contrast, the difference in foliar ethylene evolution was significant (Table 3). At 12 days after deficit irrigation, a significant difference in plant growth and hormone synthesis was detected between the small and big pot with severe drought.

**Table 3.** Effects of pot size on leaf area, transpiration, and foliar ABA content. Data are means ± standard error (SE). Different letters denote statistically significant differences between treatments ($p < 0.05$).

| Days after Deficit Irrigation | Leaf Area (cm$^2$) | | Transpiration (g m$^{-2}$ hr$^{-1}$) | | Foliar ABA (ng g$^{-1}$ DW) | | Ethylene Evolution (nL g$^{-1}$ FW) | |
|---|---|---|---|---|---|---|---|---|
| | 1.86 L | 0.94 L | 1.86 L | 0.94 L | 1.86 L | 0.94 L | 1.86 L | 0.94 L |
| 10 | 53.95 ± 4.6 [a] | 48.54 ± 3.6 [a] | 261.83 ± 21.3 [a] | 243.52 ± 23.5 [a] | 872.45 ± 67.5 [a] | 983.12 ± 56.3 [a] | 1.37 ± 0.04 [b] | 2.02 ± 0.05 [a] |
| 12 | 92.39 ± 6.3 [a] | 78.56 ± 8.1 [b] | 170.78 ± 15.2 [a] | 136.43 ± 10.5 [b] | 1474.32 ± 112.3 [b] | 2043.23 ± 151.2 [a] | 2.56 ± 0.62 [b] | 3.42 ± 0.81 [a] |

Effects of absence of canopy competition or root competition on $g_s$, ABA content, and ethylene evolution in the WT tomato under different treatments are presented in Table 4. The value of $g_s$ in the well-watered WT plants without canopy competition was significantly higher than that in the WT plant with above-and belowground competition by 25%. The influence of belowground competition on stomatal opening was insignificant under well-watered condition, as the $g_s$ between the WT tomato with and without root competition was comparable. The belowground competition, i.e., absence of canopy competition, significantly decreased the foliar ABA concentration of the competing plants, while the aboveground competition did not affect the foliar accumulation of ABA. Whereas, both the root separation and canopy separation had no remarkable effects on the concentration of ABA in the well-watered WT tomato roots. For foliar ethylene evolution of tomato, absence of canopy competition reduced the foliar ethylene evolution distinctly, but root separation had insignificant influence on foliar ethylene evolution.

**Table 4.** Abaxial stomatal conductance (mmol m$^{-2}$ s$^{-1}$), ABA content (ng g$^{-1}$ DW), and ethylene evolution (nL g$^{-1}$ FW) in competing WT plants under full irrigation and deficit irrigation. NC—without canopy competition, NR—without root competition. Data are means ± standard error (SE). Different letters denote statistically significant differences between treatments ($p < 0.05$).

| Treatment | Full Irrigation | | | | Deficit Irrigation | | | |
|---|---|---|---|---|---|---|---|---|
| | $g_s$ | Foliar ABA | Root ABA | Ethylene | $g_s$ | Foliar ABA | Root ABA | Ethylene |
| WT/WT | 238.62 ± 21.34 [b] | 601.34 ± 49.32 [a] | 49.34 ± 4.39 [a] | 1.23 ± 0.07 [a] | 144.17 ± 11.14 [a] | 1643.14 ± 137.92 [a] | 145.44 ± 14.33 [b] | 2.11 ± 0.15 [b] |
| WT/WT-NC | 298.83 ± 22.31 [a] | 561.83 ± 38.51 [b] | 48.43 ± 3.67 [a] | 0.91 ± 0.04 [b] | 143.36 ± 11.31 [a] | 1701.36 ± 132.47 [a] | 178.37 ± 13.07 [a] | 1.91 ± 0.14 [b] |
| WT/WT-NR | 247.31 ± 19.45 [b] | 611.23 ± 51.11 [a] | 51.23 ± 5.34 [a] | 1.27 ± 0.06 [a] | 157.31 ± 14.15 [a] | 1681.35 ± 138.13 [a] | 131.31 ± 15.43 [b] | 2.47 ± 0.16 [a] |

Under the condition of water deficit, the absence of aboveground competition had an indistinctive influence on the $g_s$ of the competing WT tomato, which was in contrast with the result under full irrigation. The result indicating water deficit strengthened the effects of root competition on stomatal opening. Both root separation and canopy separation did not influence the foliar concentration of ABA. In contrast to the result under full irrigation, the belowground competition increased ABA accumulation in the roots of the competing WT tomato. The non-belowground competition significantly increased the foliar ethylene evolution in the WT tomato, while the non-aboveground competition had insignificant effects on ethylene evolution.

## 4. Discussion

Intra-/inter-specific competition is the key external factor influencing plant growth, physiologies, and functions [3,36]. The few studies exploring links between plant growth traits and competition have shown that the relationships were complex, as a few plant hormones are involved in plant responses to competition [3,7,17,19,22,26]. Growing space for shoot and roots plays an important role in the interception of radiation and absorption of water and nutrients, respectively. These authors usually planted single plant and competing plants in pots with the same volume. The difference in growing space was overlooked. Unlike in the competing literature, the effect of pot size on plant response was explicitly accounted for in our experiment. Our results showed that the effects of pot size on plant growth and hormone synthesis were dependent on soil water availability. Soil drying

stimulates ABA formation in roots, translocation to leaves, then reduction in stomatal opening and plant growth [32]. Moreover, soil drying promotes soil compaction and increased plant ethylene production [37]. For the drought-stressed *flacca* tomato, ethylene produced under water stress slightly decreased leaf area, as the response of leaf growth to ethylene depends on concentration and species [15]. Our results also demonstrated that soil drying-induced compaction increased foliar ethylene evolution in both the big and small pot. Under the condition of severe drought (12 days after deficit irrigation), compared with the plants grown in big pots, the plants grown in small pots synthesized more ABA, while less ethylene. Moreover, as the physical properties were different between the compost we used in this experiment and soil, soil drying-induced compaction in soil maybe more significant. Therefore, soil compaction and soil water availability must be considered to compare soil drying-induced influences on plant physiology and phenotype in different volumes.

Plant hormones ABA and ethylene are known to regulate stomatal opening in response to competition from a neighbor [3,7,26]. With growing WT tomato and NR or FL tomato in one pot, we also concluded that ABA and ethylene are involved in plant response to intraspecific competition. The decrease in stomatal conductance, leaf area, and transpiration, induced by intraspecific competition, was accompanied by increased plant hormone concentrations. Vysotskaya et al. [7] found no significant difference in ABA concentration in xylem sap between single tomato and competing tomato. In contrast, the elevated foliar ABA concentration in competing plants was measured in our experiment, which was in accord with earlier conclusions [26,38]. Vysotskaya et al. [26] indicated that the competition from neighbors increased ABA concentration in lettuce shoot. Kurepin et al. [38] attributed the increased ABA concentration in Helianthus annuus leaves to shade light-reduced R/FR ratio. Intraspecific competition increased ABA concentration in tomato in our experiment, which contrasted with the lettuce data of Vysotskaya et al. [26]. Different species, soil, and environments may explain this.

Both ABA and ethylene are known to regulate stomatal opening in response to reduced water availability [37,39,40]. As the soil dries, the competition decreased $g_s$ by elevating ABA and ethylene accumulations. Previous studies indicated that ABA and ethylene could regulate stomatal opening antagonistically [41]. However, in the present study, both drought and intraspecific competition induced the parallel changes in ABA and ethylene synthesis, which maybe result from the inter-influence among soil water stress, soil compaction, and intraspecific competition on plant growth and hormones synthesis. This might be explained by the fact that plant response to environmental stress would depend on the ratio between concentrations of ethylene and ABA [37,42]. Under severe drought, the competition cannot reduce $g_s$ even increasing in these plant hormones accumulations. During the early stages of soil drought before hydraulic signals were produced, plant hormones dominated plant in response to stress [43]. Under severe drought, chemical signals become less important when LWP declines and leaves wilt [39]. The results indicated that the competition-induced decline in $g_s$ may be covered by the severe soil-drying-induced decrease in $g_s$, as hydraulic signals most probably dominate (Figure 2a).

Except in severe drought, competition regulated plant stomatal closure mainly through non-hydraulic signals [19,20]. Ethylene is an important phytohormone for sensing competing neighbors and determining plant responses to neighbors [7,21,22]. Under the well-watered condition, the absence of canopy competition cannot significantly reduce the stomatal opening (Table 4). Ethylene and the red:far-red light ratio (R:FR) are the most important aboveground signals of plant neighbor detection [21,44]. Canopy separation minimized the plant neighbor sensing by ethylene and R:FR. Although root exudates can serve as a belowground neighbor detection signal [45], belowground neighbor detection most probably occurs through reduction in local soil water and nutrients [46]. In the presence of aboveground sensing signals in the competing pot with root separation, the competition-induced influences on plant growth occurred observably.

Several sensing signals can detect neighbors in aboveground and belowground through neighbor-induced changes in resource availability. The absence of canopy competition significantly decreased ABA accumulation in roots and then stomatal conductance, which contrasted with the result under full irrigation, indicating the belowground neighbor detection signals were enhanced by soil drought. Although ethylene evolution was increased by drought, elevated ethylene synthesis did not significantly affect stomatal opening. Under some circumstances, ethylene can modulate stomatal responses to a given ABA concentration [17,47,48].

In this study, the influence of aboveground and belowground competition on ABA and ethylene accumulation and stomatal opening was revealed with three tomato genotypes. Association between ABA and ethylene response to intraspecific competition and drought is vital for understanding the influences of abiotic and biotic stresses on plant growth and development. However, Vysotskaya et al. [7,26] suggested that several plant hormones, such as ABA, ethylene, auxin, and cytokinins, are involved in plant growth response to competition from neighbors, while the interactive mechanism of multi-hormones regulating plant response to competition still was unclear. Further experiments are necessary to learn more about the interaction between competition and defense responses. Moreover, only a few of competition experiments were conducted under natural field conditions. Therefore, better understanding the multi-hormones mediated plant–plant interactions could help to optimize plant density and understand plant behaviors in the natural environment.

**Author Contributions:** Conceptualization, Y.G.; methodology, Y.G.; software, Y.L.; formal analysis, Y.G. and Z.S.; investigation, Y.G. and Y.F.; data curation, Y.L. and Y.F.; writing—original draft preparation, Y.G. and Z.S.; writing—review and editing, A.K.M.H.; project administration, Y.G.; funding acquisition, Y.G. All authors have read and agreed to the published version of the manuscript.

**Funding:** This research was funded by the National Natural Science Foundation of China (Grant No. 51879267) and the Agricultural Science and Technology Innovation Program (ASTIP), Chinese Academy of Agricultural Sciences.

**Institutional Review Board Statement:** Not applicable.

**Informed Consent Statement:** Not applicable.

**Data Availability Statement:** The datasets used and/or analyzed during the current study are available from the corresponding author on reasonable request.

**Acknowledgments:** We thank Ian Dodd in Lancaster University for his invaluable guidance and assistance.

**Conflicts of Interest:** The authors declare no conflict of interest.

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
