# Peer review of "Interactive Effects of Intraspecific Competition and Drought on Stomatal Conductance and Hormone Concentrations in Different Tomato Genotypes"

_horticulturae, doi:10.3390/horticulturae8010045_

Round 1
Reviewer 1 Report
In this manuscript authors have reported the effects of intraspecific competition and drought on leaf growth and stomatal conductance based on the interactive levels of ABA and ethylene. The study is well planned and the experiment was conducted with care. The manuscript however, needs a careful check for the spelling mistakes and grammar.
The major concerns in the manuscript are listed below.
- Introduction is more a general description of intraspecific description. The role of drought stress should also taken into account as one of the factors. The role of ethylene and ABA in plant growth and stomatal conductance under optimal and stressful conditions should be presented.
- Elaborate the interactive role of ethylene and ABA in regulation of stomatal conductance and stomatal distribution.
- Write on the dual role of ethylene based on the concentrations on controlling stomatal conductance.
- L82: Mention scientific name of the test plant.
- L113: Please check the measurement of leaf water potential.
- Most of the results show parallel changes in ethylene and ABA in response to competition and water availability. Please check this as increase in ethylene levels decreases ABA level.
- Elaborate the action of stress ethylene produced under water deficit and how it influence leaf area and stomatal conductance.
- L286: Please give reasons for decreased ABA and ethylene levels due to severe drought.
- More explanation is needed to show the involvement of ABA and ethylene in intraspecific competition. Few results on the hormones levels are contradictory. While discussing the concentration levels of ABA and ethylene to any response, please assure that these hormones act antagonistically. It is reported that ethylene regulates ABA for changes in stomatal conductance.
- Figures should have more resolution, a few are not clear.
- Conclude with the importance of ABA and ethylene with the objectives of the study, followed by role of other hormones as a measure of future study.
- Authors are suggested to read the following papers on ethylene/ABA and the literature cited therein.
Plants 10:180; Frontiers in Plant Science 8: 475; Plant Physiology 169:73-84
Author Response
#Reviewer 1
In this manuscript authors have reported the effects of intraspecific competition and drought on leaf growth and stomatal conductance based on the interactive levels of ABA and ethylene. The study is well planned and the experiment was conducted with care. The manuscript however, needs a careful check for the spelling mistakes and grammar.
The major concerns in the manuscript are listed below.
Comment 1: Introduction is more a general description of intraspecific description. The role of drought stress should also taken into account as one of the factors. The role of ethylene and ABA in plant growth and stomatal conductance under optimal and stressful conditions should be presented.
Response: Thanks for your suggestions. The section of Introduction was revised according to your suggestions. (Lines 46-57)
Comment 2: Elaborate the interactive role of ethylene and ABA in regulation of stomatal conductance and stomatal distribution.
Response: Thank you. The interactive role of ethylene and ABA in regulation of stomatal conductance and stomatal distribution was present. (Lines 55-57, Lines 344-345)
Comment 3: Write on the dual role of ethylene based on the concentrations on controlling stomatal conductance.
Response: Thanks. The dual role of ethylene based on the concentrations on controlling stomatal conductance was present. (Lines 53-54)
Comment 4: L82: Mention scientific name of the test plant.
Response: The scientific name of tomato, Solanum lycopersicum, was added in Line 96.
Comment 5: L113: Please check the measurement of leaf water potential.
Response: There was a mistake in the measurement of leaf water potential. We have modified the measurement of leaf water potential of tomato plants (Lines 126-128).
Comment 6: Most of the results show parallel changes in ethylene and ABA in response to competition and water availability. Please check this as increase in ethylene levels decreases ABA level.
Response: Thanks for your suggestions. We checked the data of ABA and ethylene. The parallel changes in ethylene and ABA may be result from the inter-influence among soil water stress, soil compaction, and intraspecific com-petition on plant growth and hormones synthesis. (Lines 345-349)
Comment 7: Elaborate the action of stress ethylene produced under water deficit and how it influence leaf area and stomatal conductance.
Response: Thanks. The effect of ethylene produced under water stress on tomato plants were added. (Lines 317-319)
Comment 8: L286: Please give reasons for decreased ABA and ethylene levels due to severe drought.
Response: There was a mistake. The sentence was revised as: Under the condition of severe drought (12 days after deficit irrigation), compared with the plants grown in big pots, the plants grown in small pots synthesized more ABA, while less ethylene. (Line 321-323)
Comment 9: More explanation is needed to show the involvement of ABA and ethylene in intraspecific competition. Few results on the hormones levels are contradictory. While discussing the concentration levels of ABA and ethylene to any response, please assure that these hormones act antagonistically. It is reported that ethylene regulates ABA for changes in stomatal conductance.
Response: Thank you. According to your suggestions, we revised the discussion on ABA and ethylene. (Lines 344-350)
Comment 10: Figures should have more resolution, a few are not clear.
Response: Thanks. Resolution of figures were improved.
Comment 11: Conclude with the importance of ABA and ethylene with the objectives of the study, followed by role of other hormones as a measure of future study.
Response: Thank you. The importance of ABA and ethylene with the objectives of the study was conclude, and the role of other hormones as a measure of future study was present at the end of Discussion. (Lines 378-388)
Comment 12: Authors are suggested to read the following papers on ethylene/ABA and the literature cited therein. Plants 10:180; Frontiers in Plant Science 8: 475; Plant Physiology 169:73-84.
Response: Thanks. The three papers were cited in the paper.

Reviewer 2 Report
The aim of this study is to evaluate effect of intraspecific competition and water regime on physiological state of tomato. 3 tomato genotypes were used in this study. I like the idea and design of this study; however, I have some minor suggestions to authors:
Table 2 should mention what brackets mean, i. e. standard error, standard deviation, give confidence level or other. Some information is presented in Materials and Methods, however, each table or figure must be understood without looking elsewhere.
My main suggestion is to formulate clear conclusion at the end of discussion, prospects for future studies is not enough. Authors must present their findings in research paper.
Figure 1 must go to Results section.
I think English should be corrected as well.
In general, I think manuscript should be accepted after minor correction.
Author Response
# Reviewer 2
The aim of this study is to evaluate effect of intraspecific competition and water regime on physiological state of tomato. 3 tomato genotypes were used in this study. I like the idea and design of this study; however, I have some minor suggestions to authors:
Comment 1: Table 2 should mention what brackets mean, i. e. standard error, standard deviation, give confidence level or other. Some information is presented in Materials and Methods, however, each table or figure must be understood without looking elsewhere.
Response: Thanks for your suggestions. For Table 2, 3, and 4, data are means ± standard error (SE), and different letters denote statistically significant differences between treatments (P < 0.05).
Comment 2: My main suggestion is to formulate clear conclusion at the end of discussion, prospects for future studies is not enough. Authors must present their findings in research paper.
Response: Thanks for your suggestions. The findings of this study were present at the end of Discussion. (Lines 376-378)
Comment 3: Figure 1 must go to Results section.
Response: Figure 1 was moved to the section of Results.
Comment 4: I think English should be corrected as well.
Response: English of this manuscript was polished by a native English speaker.
Comment 5: In general, I think manuscript should be accepted after minor correction.
Response: Thanks for your appreciation.

Round 2
Reviewer 1 Report
The readability of the manuscript has improved appropriately. Authors have addressed the concerns raised on the earlier version satisfactorily.